# Theory and Applications of the (Cardio) Genomic Fabric Approach to Post-Ischemic and Hypoxia-Induced Heart Failure

**DOI:** 10.3390/jpm12081246

**Published:** 2022-07-29

**Authors:** Dumitru Andrei Iacobas, Lei Xi

**Affiliations:** 1Personalized Genomics Laboratory, Roy G Perry College of Engineering, Prairie View A&M University, Prairie View, TX 77446, USA; 2Pauley Heart Center, Division of Cardiology, Virginia Commonwealth University, Richmond, VA 23298, USA

**Keywords:** *Adra1b*, *Ank2*, bone marrow stem cell therapy, *Cox6b1*, crem, gene expression control, gene expression coordination, lias, transcriptomic distance, transcriptomic stoichiometry

## Abstract

The genomic fabric paradigm (GFP) characterizes the transcriptome topology by the transcripts’ abundances, the variability of the expression profile, and the inter-coordination of gene expressions in each pathophysiological condition. The expression variability analysis provides an indirect estimate of the cell capability to limit the stochastic fluctuations of the expression levels of key genes, while the expression coordination analysis determines the gene networks in functional pathways. This report illustrates the theoretical bases and the mathematical framework of the GFP with applications to our microarray data from mouse models of post ischemic, and constant and intermittent hypoxia-induced heart failures. GFP analyses revealed the myocardium priorities in keeping the expression of key genes within narrow intervals, determined the statistically significant gene interlinkages, and identified the gene master regulators in the mouse heart left ventricle under normal and ischemic conditions. We quantified the expression regulation, alteration of the expression control, and remodeling of the gene networks caused by the oxygen deprivation and determined the efficacy of the bone marrow mono-nuclear stem cell injections to restore the normal transcriptome. Through the comprehensive assessment of the transcriptome, GFP would pave the way towards the development of personalized gene therapy of cardiac diseases.

## 1. Introduction

Personalized (or precision medicine) has become a focal area of interest and development in medicine of the 21st century. This is because the individual variances of biological response to environmental factors and drug therapies may substantially alter the outcomes of disease progression and therapeutic success. Understanding the genetic and epigenetic mechanisms regulating these variabilities due to the factors, such as race, sex, age, medical history, diet, stress, exposure to toxins, and individual habits, etc., could be critical for designing a clinical strategy tailored for each patient. For example, 70 microRNAs identified in whole blood samples via next generation sequencing were linked to the risk of recurrent myocardial infarction and future stent thrombosis, ascompared to coronary artery disease (CAD) patients without the subsequent events [1]. A recent review elegantly summarized advances that unravel the genetic architecture of CAD with approximately 60 genetic loci to CAD risk [2]. These authors suggested that genetic testing could enable precision medicine approaches by identifying subgroups of patients at an increased risk of CAD or with a specific driving pathophysiology that can be targeted precisely [2].

Gene expression profiling is a very important tool to individualize the molecular mechanisms responsible for the pathophysiological characteristics and tailor the appropriate therapy for each individual (e.g., [3,4,5]), but do we utilize the expression data at their full potential? As we will prove in this report, traditional quantification of the transcriptomic alteration, limited to identify the up/down-regulated and turned on/off genes, neglects over 99% of the information provided by any high-throughput gene expression platform. Therefore, we use the genomic fabric paradigm (GFP) [6] that, in addition to the average expression level (AVE), characterizes each individual gene by the relative expression variability (REV) across biological replicas and the expression correlation (COR) with each and other genes in the same condition.

REV is an indirect measure of the strength of the homeostatic mechanisms that ensure the right expression of key genes by limiting their random fluctuations caused by the stochasticity of the chemical reactions and environment unsteadiness within narrow intervals. On the other hand, COR, accounting for the “Principle of Transcriptomic Stoichiometry” (PTS) [7], determines whether and how strongly the genes are networked in functional pathways. PTS, a generalization of the Proust’s law of constant proportion (definite composition) [8] from chemistry, requires the coordinated expression of genes networked in functional pathways. GFP considers the transcriptome as a multi-dimensional mathematical object subjected to dynamic sets of homeostatic controlling mechanisms and expression correlations among the individual genes. 

We have shown previously that the transcriptome topology is strongly dependent on race/strain [9], sex [10], age [11], and region of the profiled tissue [6]. In addition to these general factors, the transcriptomic organization is also very sensitive to the individual’s chronic or/and acute diseases [12], diet [13], medical treatments [14], external stimuli [15], and a wide diversity of habits (smocking, alcohol, and drugs), exposure to stress [16], and infections [17,18]. As subjected to unrepeatable combinations of influential factors, some of them changing in time, each human is dynamically unique and therefore the medical treatment should be tailored to his/her actual characteristics. This report presents the theoretical bases of the (cardio) genomic fabric approach, an important step towards development of the personalized cardiology. In order to fluidize the reading, previously published mathematical formulae were included in Appendix B.

## 2. Materials and Methods

### 2.1. Experimental Data

The (cardio)genomic fabric approach is illustrated here by using publicly available microarray data from a mouse model of post-ischemic heart failure (PIHF) [19,20] and from mice subjected to chronic constant (CCH) or intermittent (CIH) hypoxia during their first 1, 2, or 4 weeks of life [21]. In these studies, the gene expressions were profiled in the left ventricle of each of the four mice in every experimental group using 32k mouse oligonucleotide microarrays printed by the Microarray Facility of the Albert Einstein College of Medicine, Bronx, NY, USA [22].

As described in Ref. [12], the myocardial infarction was induced into anesthetized adult C57BL/6 mice by ligating the descending branch of the left coronary artery. Mice with developed PIHF were then injected into 3 regions at the borders of the cardiac scar with 10 µL Matrigel (BD Biosciences) with or without 1.5 × 10^6^ bone marrow mononuclear stem cells. The transcriptomes were profiled 59 days after induction of myocardial infarction and 49 days after cell therapy. Total 12 mice (n = 4 per group) were used in this experiment and divided into: normal untreated (“NN”), infarcted untreated (“IN”), and infarcted treated (“IT”). As mentioned in Refs. [12,14], the experiments respected the Guide for the Care and Use of Laboratory Animals and were approved by the Animal Committee of Universidade Federal do Rio de Janeiro, Rio de Janeiro, RJ, Brazil.

In the hypoxia study, neonatal CD1 mice were placed in Biospherix hypoxia chambers in the second day of their life and kept there for 1, 2, or 4 weeks. Three groups of 4 mice each, denoted by “N1”, “N2”, “N4”, were kept under normal atmospheric conditions (F_i_O_2_ = 21%). For three groups of 4 mice each, denoted by “I1”, “I2”, “I4”, F_i_O_2_ was alternated between 21% and 11% every 4 min, 24 h/day 1, 2 or 4 weeks. Finally, for other three groups of 4 mice each, denoted by “C1”, “C2”, “C4”, F_i_O_2_ was kept constant at 11% for the entire period [13] of 1, 2, or 4 weeks. CIH experiment modeled the sleep apnea, while CCH experiment modeled the living at high altitude. As mentioned in Ref. [16], the experimental investigations were approved by the Institutional Animal Care and Use (IACUC) of the Albert Einstein College of Medicine, Bronx, NY, USA.

In all experiments, every group of mice was composed of two males and two females, preferably from the same litter. Nonetheless, with only two males and two females per group, the sex differences were not statistically significant. Although the datasets from these experiments were presented in previous publications [12,14,16,23,24,25], the (cardio) genomic fabric approach was never used at its full potential. As shown in this report, this approach is able to reveal novel features for which the traditional analysis is not equipped to delineate. 

### 2.2. Characteristics of the (Cardio) Genomic Fabric 

We define the (cardio) genomic fabric of a functional pathway in a particular region of myocardium as the transcriptome associated to the most interconnected and stably expressed gene network responsible for that pathway in that heart region. 

In each condition, the independent characteristics AVE, REV, and COR of every gene were averaged over the valid spots probing that gene ([26], Appendix B, Equations (A1)–(A3). Our normalization procedure returns the AVE values in terms of the median gene expression level in each condition. For instance, AVE = 4.31 for prolactin receptor, (*Prlr*) in N1 means that the average expression of this gene is 4.31 × larger than that of the median gene in N1 (like *Slc44a2*-Solute carrier family 44, member 2 whose AVE = 1.0005).

REV was computed as the mid-interval of the chi-square estimate of the expression level coefficient of variation among biological replicas. It was further used to determine the relative expression control (REC) of individual genes and the pathway relative expression control (PREC) of the (cardio) genomic fabrics: (1)$$\begin{array}{l} REC_{i}^{\left(condition\right)}=\frac{{\langle REV_{i}^{\left(condition\right)}\rangle |}_{\mathrm{all}\;\mathrm{genes}}}{REV_{i}^{\left(condition\right)}}-1\;,\;where:\;\\ {\langle REV_{i}^{\left(condition\right)}\rangle |}_{\mathrm{all}\;\mathrm{genes}}=\mathrm{median}\;\mathrm{REV}\;\mathrm{over}\;\mathrm{the}\;\mathrm{entire}\;\mathrm{transcriptome} \end{array}$$
(2)$$\begin{array}{l} PREC_{\Gamma }^{\left(condition\right)}=\frac{{\langle REV_{i}^{\left(condition\right)}\rangle |}_{\mathrm{all}\;\mathrm{genes}}}{{\langle REV_{i}^{\left(condition\right)}\rangle |}_{i\in \Gamma }}-1\\ where:\;{\langle REV_{i}^{\left(condition\right)}\rangle |}_{\Gamma }=\mathrm{median}\;\mathrm{REV}\;\mathrm{over}\;\mathrm{the}\;\mathrm{pathway}\;\Gamma \; \end{array}$$

Higher positive REC values indicate genes whose random fluctuations of the expression level are strongly limited by the cellular homeostatic mechanisms within narrow intervals as their right expressions are critical for the cardiac physiology. By contrast, large negative RECs are associated with less controlled genes useful for cell adaptation to environmental changes. Thus, REC indicates the priorities of the cell in regulating the transcription machinery. Similarly, high positive PRECs are associated with critically important pathways for the preservation of the phenotypic expression fluctuations and negative PRECs with adapting pathways. One may observe that REC = 0 set the baseline for genes and PREC = 0 the baseline for the pathways. As presented in the Results section below, REC and PREC are sensitive to the external factors such as ischemia and oxygen deprivation.

COR is the Pearson product-moment correlation coefficient between the (log_2_) expressions of two genes across biological replicas ([26], Appendix B). COR analysis identifies the (*p*-value < 0.05) significantly synergistically, antagonistically, and independently expressed gene pairs, albeit it cannot determine which of the paired genes is the master. Thus, COR determines the statistically significant gene networks in each condition, refining the gene “wiring” in functional pathways constructed by dedicated software such as: QIAGEN Ingenuity Pathway Analysis [27], DAVID [28], KEGG [29], etc. Without COR analysis, the traditional pathways do not account for individual factors known to influence the incidence of the disease and the response to a treatment. 

Moreover, GFP establishes the gene hierarchy in each condition using the gene commanding height (GCH) scoring that combines the expression control and expression coordination with each and other genes [26]. The top of the hierarchy (highest GCH, termed gene master regulator, GMR) is the gene whose strongly protected expression level is the most influential on the expression and networking of other genes.
(3)$$GCH_{i}^{\left(condition\right)}=\left(REC_{i}^{\left(condition\right)}+1\right)\times \exp\left(4\underset{\mathrm{average}\;\mathrm{of}\;\mathrm{squares}}{\underset{︸}{\bar{{COR_{ig}^{2}|}_{\forall g\neq i}^{\left(condition\right)}}}}-1\right)$$

By stably transfecting two human thyroid cancer cell lines with four genes, we proved that expression manipulation of a gene has transcriptomic consequences proportional to the GCH of that gene [30]. Because each cell phenotype has distinct gene hierarchy, smart manipulation of the GMR expression can be used to selectively kill, or by contrary, stimulate the proliferation of the desired cell type from a tissue [31].

### 2.3. Comparing Conditions of the (Cardio) Genomic Fabric

#### 2.3.1. Cut-Off Criteria

When comparing two conditions (for instance “IN” with “NN” in the infarct experiment), traditional transcriptomic analysis uses uniform, arbitrarily introduced cut-off for the absolute fold-change (1.5× or 2.0×) of the expression ratio “*x*” (negative for down-regulation). Some analyses require also a less than 0.05 *p*-value of the heteroscedastic *t*-test of the two means equality. However, such absolute fold-change cut-off might be too stringent for very stably expressed genes across biological replicas and low local technical noise, while for other genes it might be too relaxed. Therefore, while maintaining the *p*-value < 0.05 condition, we determine the absolute fold-change cut-off for the expression ratio “*x*” separately for each quantified gene to account for both biological variability and the technical noise of the probing spot(s) ([26], Appendix B, Equation (A4)). 

#### 2.3.2. Uniform, Weighted, and All-Inclusive Contributions of Individual Genes to the Transcriptome Alteration

Traditional analysis measures the overall transcriptomic alteration by the percentages of the genes that were significantly up-/down-regulated or turned on/off. For this measure, the significantly regulated genes are considered as uniform +1 or −1 contributors to the transcriptome alteration. 

A more informative measure is the **We**ighted **I**ndividual (gene) **R**egulation (WIR) [32] that takes into account the absolute departure from the normal expression level and the statistical confidence in the expression regulation. Like the expression ratio “*x*”, WIR takes also positive values for up-regulated genes and negative values for the down-regulated ones ([26], Appendix B, Equation (A5)).

However, the best all-inclusive characterization of one gene contribution to the overall transcriptomic alteration is the “**I**ndividual (gene) **T**ranscriptomic **D**istance” (ITD). ITD is the magnitude of the 3D vector whose orthogonal components reflect the relative changes in the average expression level, expression variability (among biological replicas), and expression correlations (averaged over all other expressed genes) ([33], Appendix B, Equation (A6)).

Both WIR and ITD can be further averaged for a given functional pathway “Γ” as the weighted pathway regulation (WPR) and **P**athway **T**ranscriptomic (**t**rajectory) **D**istance (PTD), much more accurate in ranking the pathways according to their alteration than the percentage of regulated genes.
(4)$$\begin{array}{l} WPR_{\Gamma }^{\left(A\to B\right)}=\frac{100}{\left\{\Gamma \right\}}\sqrt{\sum_{i\in \Gamma }{\left(WIR_{i}^{\left(A\to B\right)}\right)}^{2}}\;,\;\{\Gamma \}=\;\mathrm{number}\;\mathrm{of}\;\mathrm{genes}\;\mathrm{in}\;\Gamma \\ PTD_{\Gamma }^{\left(A\to B\right)}=\frac{100}{\left\{\Gamma \right\}}\sqrt{\sum_{i\in \Gamma }{\left(ITD_{i}^{\left(A\to B\right)}\right)}^{2}} \end{array}$$

### 2.4. Transcriptomic Effect of a Treatment

One can determine the transcriptomic effect of a treatment by comparing the alterations before and after that treatment. The comparison may encompass all genes or may be restricted to a particular functional pathway “Γ”. Traditionally, such comparison is done by comparing the numbers of regulated genes with and without the treatment (when each gene is considered as a uniform contributor). 

In previous papers [14,34,35], we used the **G**ene **E**xpression **R**ecovery (GER, Appendix B, Equation (A7)). GER takes into account not only the numbers of up-(U) and down-(D) regulated genes in “IN” whose normal expression was fully recovered (X) in IT (i.e., {DX} and {UX}), but also the genes whose regulation status was not changed (i.e., {DD} and {UU}) and those whose regulation type was switched in IT (i.e., {DU} and (UD}).

A better measure of the transcriptomic restoration compares the WPR scores, as the **P**athway **R**estoration **E**fficiency (PRE) ([35], Appendix B, Equation (A8)).

Now, we add the **C**omprehensive **P**athway **R**estoration (CPR) representing the percent reduction of the transcriptomic distance in response to the treatment.
(5)$$\begin{array}{l} CPR_{\Gamma }^{\left(IN\to IT;NN\right)}=\left(1-\frac{PTD_{\Gamma }^{\left(NN\to IT\right)}}{PTD_{\Gamma }^{\left(NN\to IN\right)}}\right)\times 100\%\\ \mathrm{Possible}\;\mathrm{outcomes}:\\ \begin{array}{llll} \left(a\right) & PTD_{\Gamma }^{\left(NN\to IT\right)}=0\Rightarrow CPR_{\Gamma }^{\left(IN\to IT;NN\right)}=100\% & , & ideal\\ \left(b\right) & 0<PTD_{\Gamma }^{\left(NN\to IT\right)}<PTD_{\Gamma }^{\left(NN\to IN\right)}\Rightarrow 0<CPR_{\Gamma }^{\left(IN\to IT;NN\right)}<100\% & , & positive\\ \left(c\right) & PTD_{\Gamma }^{\left(NN\to IT\right)}=PTD_{\Gamma }^{\left(NN\to IN\right)}\Rightarrow CPR_{\Gamma }^{\left(IN\to IT;NN\right)}=0\% & , & null\\ \left(d\right) & PTD_{\Gamma }^{\left(NN\to IT\right)}>PTD_{\Gamma }^{\left(NN\to IN\right)}\Rightarrow CPR_{\Gamma }^{\left(IN\to IT;NN\right)}<0\% & , & negative \end{array} \end{array}$$

## 3. Results

### 3.1. Overview of the Microarray Data 

The PIHF experiment quantified the expression levels of 10,408 unigenes in all 12 samples from the groups “NN”, “IN”, “IT”, n = 4/group), from which the GFP was extracted in each condition 10,408 AVEs + 10,408 REVs + 54,158,028 CORs = 54,178,844 values. Thus, the GFP increased the size of the workable data in each condition by 5206 times. With respect to “NN”, 579 genes (i.e., 5.56%) were significantly (according to our composite criterion) up-regulated and 1222 (11.74%) were down-regulated in “IN”. The treatment with bone marrow mononuclear stem cells partially recovered the normal expression levels of the genes, leaving 256 (2.46%) up-regulated and 667 (6.41%) down-regulated in the “IT” group as compared with the “NN” controls. Interestingly, the treatment went even further by flipping the significant down-regulation of 19 genes and the significant up-regulation of 15 genes to their opposites. 

In the hypoxia experiment, 9716 unigenes were adequately quantified in all 36 samples from the groups “N1”, “I1”, “C1”, “N2”, “I2”, “C2”, “N4”, “I4”, “C4” (n = 4/group), from which GFP extracted in each condition 47,214,902 values. This is an increase by 4860 times of the information used in the traditional analysis limited to the AVEs. According to our criterion, with respect to the control group “N1”, 9.43% of the genes were up- and 11.72% were down-regulated in the group “I1”, and 6.13% were up- and 4.30% down-regulated in “C1”. With respect to “N2”, 10.48% of the genes were up- and 18.94% were down-regulated in “I2”, and 22.02% up- and 18.92% down-regulated in “C2”. Finally, with respect to “N4”, 4.08% were up- and 2.14% down-regulated in “I4”, while in “C4”, 6.01% were up- and 6.71% down-regulated.

The genes with the largest expression level were: *Hspb6* (heat shock protein, alpha-crystallin-related, B6; AVE = 14.19 in “N1”, 16.51 in “I2”, and 14.12 in “C2”; and *Nr1i3* (nuclear receptor subfamily 1, group I, member 3; with AVE = 15.05 in “I1”, 24.96 in “N4”, 18.97 in “I4”, and 21.06 in “C4”. *Hspb6*, encoding a small heat shock protein related to oxidative stress, is an important modulator of muscle contraction [36]. *Nr1i3*, also known as constitutive androstane receptor (*Cas*), is an important modulator of energy pathways and xenobiotic metabolism [37].

### 3.2. AVE, REV, and COR Are Independent Features

Figure 1 illustrates the independence of AVE, REV, and COR (with *Ank2*-ankyrin 2) for 34 inflammatory response genes in the heart left ventricle of mice subjected for the first week of their life to normal atmospheric conditions (“N1”), intermittent hypoxia (“I1”), or constant hypoxia (“C1”). Appendix A present the AVEs, REVs, and CORs with *Ank2* of the same genes for the 2 and 4 weeks of hypoxia exposures. The value 1 for the expression correlation of *Ank2* with itself is the validation of the correctness of the COR analysis. Although Figure 1 is restricted to this subset of genes, any other subset of genes in any other condition would prove the independence, including the ion channels and transporters in each of the four heart chambers of the adult mouse [6].

Figure 1 and Appendix A also show that chronic hypoxia not only changed the average expression level, but also the homeostatic control of the expression fluctuations and, hence the expression variability across biological replicas. On top of this, hypoxia changed the expression coordination with other genes, illustrated here with *Ank2*, encoding Ankyrin-B, one major player in cardiac physiology [38]). Coordination changes indicate remodeling of the gene networks. Through the analyses of expression variability and expression coordination, the GFP brings a treasure of (previously neglected) information about how much the cardiac transcriptome is controlled and organized in partially overlapping networks. Our data indicate that changes in expression variability and expression coordination are not negligible and should be considered when quantifying the overall transcriptomic alteration in a disease and recovery following a treatment.

### 3.3. Control of Gene Expression

Table 1 lists the five genes with the strongest expression control (i.e., the lowest REV) in each group of animals from the hypoxia experiments. Of note is the diversity of the most controlled genes, with no overlap among sets until the 15th most controlled genes. It is very interesting that the left ventricle prioritized the control of different genes at each of the three developmental stages that chronic hypoxia altered these priorities, and that constant and intermittent oxygen deprivation had discrepant effects. We observed also (and presented in the last two rows of the table) that both the average and the median REVs for all quantified genes are larger at normal atmospheric conditions than in CIH and CCH at all three ages. 

### 3.4. Gene Hierarchy

GCH analysis was used to establish the hierarchies of the genes in the conditions “NN”, “IN”, and “IT”. Figure 2 presents the top 20 genes in each condition of the post-ischemic cardiac failure experiment. Of note is the lack of overlap among the three sets of the top 20 genes, indicating distinct transcriptomic topologies. Remarkably, the top genes in one condition have low GCH scores in the other two. This finding can be used to selectively target the cells commanded by the GMRs in a heterocellular tissue (like in a prostate cancer tumor harboring both cancer and normal cells, [26]).

For the conditions analyzed here, the gene master regulators (GMRs–top GCH scorers) are: transmembrane protein 186 (*Tmem186*, with GCH = 53.21 in “NN”, 1.03 in “IN”, and 2.57 in “IT”), CD164 antigen (*Cd164*, with GCH = 1.54 in “NN”, 46.37 in “IN”, and 2.08 in “IT”), and ATPase type 13A2 (*Atp13a2*, with GCH = 1.88 in “NN”, 0.99 in “IN”, and 32.43 in “IT”).

*Tmem186* top position in the heart of adult healthy mice may be related to its role in the mitochondrial complex I [39], while the 5th position of *Tmem208* by its regulatory function in autophagy [40]. Existing literature justifies why some of the top hits in the above GCH analysis might be considered as potential new gene targets to protect the heart against ischemic injury. For example, under the “IN” condition, top genes include regulators of the cellular metabolism, suggesting adaptive responses to myocardial infarction. Most notably, the No. 2 gene hit- *Cox6b1* (cytochrome c oxidase, subunit 6B1) was shown to protect cardiomyocytes from hypoxia/reoxygenation injury by reducing the production of reactive oxygen species and cell apoptosis [41]. The 4th gene hit in “IN”-*Pcsk7* (proprotein convertase subtilisin/kexin type 7) was also associated with cardiovascular disease phenotypes [42]. On the other hand, under IT condition, the 6th top hit gene- *Tuba1c* (tubulin, alpha 1C) was previously identified to predict the outcome of a linear combination of circadian rhythm pathway genes [43]. The abundance of the encoded protein by the 8th top gene, *Fam171a2* (family with sequence similarity 171, member A2) in the heart was recently correlated with the PR interval of an electrocardiogram, suggesting a role for the cardiac conduction system [44]. Thus, the GCH analysis can be an important tool to identify novel therapeutic avenues for cardiac diseases.

### 3.5. Measures of Expression Regulation

Figure 3 illustrates, for 40 genes involved in the adrenergic signaling in cardiomyocytes [45], the four ways to report their transcriptomic alteration in infarcted untreated (“IN”) and treated (“IT”) hearts with respect to the normal condition.

In the traditional analysis of the percentages of significantly up-/down-regulated genes, each affected gene is considered as a uniform +1 or −1 contributor to the overall transcriptomic regulation. Moreover, this measure is limited to the significantly regulated genes according to the criterion established by the investigator, frequently an arbitrary absolute fold-change cut-off. In our study, the absolute fold-change cut-off is determined separately for each expressed gene pending on its expression variability across biological replicas and the technical noise of the probing spot(s) in the microarray [26].

A better way to quantify the contributions of the individual genes to the overall transcriptomic alteration is to use the WIR score. Although still limited to the change in the expression level, WIR not only considers all genes but weights their contribution according to the total absolute change of their expression level and the statistical confidence in their regulation. This measure was previously used to quantify the transcriptomic alterations in the left ventricle of a mouse model of Chagasic cardiomyopathy [32], in the cortical oligodendrocytes and microglia of a rabbit model of intra-ventricular hemorrhage [46], and in the hypothalamic arcuate node of a rat model of infantile spasms [35].

Nevertheless, the most comprehensive measure is the ITD that takes into account the alterations of all independent characteristics of the individual genes. The new measures reveal that genes neglected because their expression ratios did not pass the threshold to be considered as significantly regulated may still have contributions to the transcriptome alteration through affected expression control mechanisms and remodeling of the gene networks. For instance, although the expression ratio of the significantly up-regulated *Tpm1* (Tropomyosin 1, alpha) in “IN” (2.18×) is larger than that of the not significantly regulated *Tpm4* (Tropomyosin 4, 1.31×), both WIR and ITD scores are larger for Tpm4 (WIR = 0.68, ITD = 1.89) than for *Tpm1* (WIR = 0.55, ITD = 0.30) in “IN”. Such findings impose a reconsideration of what really matters in the transcriptome changes. Interestingly, while *Tpm1* is one of the main hypertrophic cardiomyopathy genes [47], Tpm4 is known for its inhibitory effect on actin polymerization [48].

Both “WIR” and “ITD” analyses revealed that for this pathway, the regulation of the *Gnas* gene, encoding the stimulatory alpha subunit of the protein complex guanine nucleotide-binding protein (G protein), had the largest contribution to the transcriptomic alteration in “IN” (WIR = −27.98, ITD = 4.47). Although, no longer significantly regulated in “IT” (x = −1.12), *Gnas* still contributes to the transcriptomic differences with respect to the control “NN” (WIR = −1.67, ITD = 2.86). It was recently reported that a somatic mutation of *Gnas* is associated with focal, idiopathic right ventricular outflow tract (RVOT) tachycardia [49].

### 3.6. Regulation of the Adrenergic Signaling in the Left Ventricle of Mice with Post-Ischemic Heart Failure

Figure 4 presents the significant regulation of the genes involved in the adrenergic signaling in cardiomyocytes [45] as indicated by the microarray data in the left ventricle of mice with post-ischemic heart failure (condition “IN” with respect to “NN”). The pathway was designed by the Kanehisa Laboratories who developed the Kyoto Encyclopedia of Genes and Genomes (KEGG, [50]).

Thirteen genes of this pathway were significantly regulated by the infarct: *Adra1b* (adrenergic receptor, alpha 1b), *Atf2* (activating transcription factor 2), *Atp1b2* (ATPase, Na^+^/K^+^ transporting, beta 2 polypeptide), *Bcl2* (B-cell leukemia/lymphoma 2), *Cacnb2* (calcium channel, voltage-dependent, beta 2 subunit), *Calm2* (calmodulin 2), *Fxyd2* (FXYD domain-containing ion transport regulator 2), *Gnas* (guanine nucleotide binding protein, alpha stimulating) complex locus), *Kcne1* (potassium voltage-gated channel, Isk-related subfamily, member 1), *Ppp1cc* (protein phosphatase 1, catalytic subunit, gamma isoform), *Ppp2r2d* (protein phosphatase 2, regulatory subunit B, delta isoform), *Rapgef4* (rap guanine nucleotide exchange factor (GEF) 4), *Scn7a* (sodium channel, voltage-gated, type VII, alpha), and *Tpm1* (tropomyosin 1, alpha).

One may note that *Adra1b*, involved in the positive regulation of the blood pressure [51], was significantly down-regulated (x = −1.81, WIR = −1.72, ITD = 1.50) in the infarcted heart. Although the other two subtypes of the alpha-adrenergic receptors [52] were also down-regulated (*Adra1a*: x = −1.19, CUT = 1.65; *Adra1d*: x = −1.44, CUT = 1.73), their regulations were not statistically significant. However, because of larger differences in REVs and CORs, both genes had higher contributions to the overall alteration of the transcriptome than *Adra1b*, with ITD = 1.62 (*Adra1a*) and ITD = 1.67 (*Adra1d*). It was reported that the stimulation of these alpha-adrenergic receptors can protect cardiomyocytes against ischemia by regulating the influx of glucose [53].

### 3.7. Recovery of the Adrenergic Signaling in the Left Ventricle of Mice with Post-Ischemic Heart Failure following Treatment with Bone Marrow Mononuclear Stem Cells

Figure 5 presents the regulation of the adrenergic signaling in cardiomyocyte pathways after the stem cell treatment (condition “IT” with respect to “NN”). Of note is the recovery of the normal expression for: *Adra1b*, *Atp1b2*, *Bcl2*, *Cacnb2*, *Calm2*, *Fxyd2*, *Gnas*, *Ppp1cc*, *Ppp2r2d*, *Rapgef4*, *Scn7a*, and *Tpm1*. *Kcne1* remained down-regulated, while the normally expressed *Cacna2d1* (calcium channel, voltage-dependent, alpha2/delta subunit 1) and *Ppp2r5e* (protein phosphatase 2, regulatory subunit B (B56), epsilon isoform) in “IN” condition were down-regulated by the treatment. Therefore, for this pathway, {DX} = 7, {UX} = 5, {XD} = 2, {XU} = 0, {DD} =1, {UU} = 0, {UD} = 0, {DU} = 0, so that GER^(IN^^→IT;NN)^ = 66.67%. WPR^(NN^^→IN)^ = 36.10 and WPR^(NN^^→IT)^ = 13.63, making WPR^(IN^^→IT;NN)^ = 62.24%. The transcriptomic distance analysis returned: PTD^(NN^^→IN)^ = 37.11, PTD^(NN^^→IT)^ = 11.29, resulting CPR^(IN^^→IT;NN)^ = 69.59%.

### 3.8. Reconfiguration of the Gene Networks by the Post-Ischemic Heart Failure and Recovery following Treatment with Bone Marrow Mononuclear Stem Cells

Figure 6 illustrates the reconfiguration of the gene networks by the disease and following a treatment by showing the changes in the expression correlations of *Adra1b* with the other genes from the adrenergic signaling in cardiomyocytes. *Adra1b*, one of the six subtypes of the adrenergic receptors that control the heart contractility (inotropism) and rate (chronotropism), mediates its action by association with G proteins that activate a phosphatidylinositol–calcium second messenger system. In mouse, the alpha1-adrenergic receptors play adaptive roles in the heart and protect against the development of heart failure [54]. Figure 6a–c present only the genes of the pathway that are statistically (*p* < 0.05) significant synergistically or antagonistically expressed with *Adra1b* in at least one of the three conditions. However, the percentages of the synergistically, antagonistically, and independently expressed partners were computed for the entire pathway. Figure 6d lists the genes that are independently expressed with *Adra1b* in each condition.

Interestingly, as shown in panel (b), the infarct increased the synergistic partnership of *Adra1b* in this pathway from 13.9% to 24.1% and that of the antagonistic partnership from 1.3% to 8.9%. This very substantial strengthening of the *Adra1b* inter-coordination with many other genes of the pathway makes *Adra1b* a very important target to recover the altered heart functions. The treatment (panel (c)), reduced back the expression coordination to 11.4% synergism and 1.3% antagonism, and further decoupled numerous other genes from their correlation with *Adra1b* (panel (d): *Akt1* (thymoma viral proto-oncogene 1), *Atf2*, *Atf4* (activating transcription factor 4), *Atp1a3* (ATPase, Na^+^/K^+^ transporting, alpha 3 polypeptide), *Cacnb2*, *Camk2a*, *Ppp2r2d*, and *Ppp2r5d* (protein phosphatase 2, regulatory subunit B (B56), delta isoform). However, the treatment antagonistically coupled the independently expressed *Atf6b* (activating transcription factor 6 beta) and *Adra1b* under ischemic conditions.

## 4. Discussion

The present study provides the theoretical bases of the (cardio)genomic fabric approach in identifying key gene regulatory factors. The theory is applied to expression data from the hearts of mouse models of common myocardial pathologies such as hypoxia and myocardial infarction with or without treatment with bone marrow mononuclear stem cells. Nevertheless, in a really personalized application, the biological replicas of diseased and normal cells should come from the same individual. Such a procedure has been already used to compare samples collected from each cancer nodule and surrounding normal tissue from surgically removed tumors of the thyroid [30,31], kidney [33], and prostate [26,55]. For cardiac diseases, one can collect samples from localized heart myocardium using percutaneous endomyocardial biopsy [56]. Although the experimental results were intended to illustrate how the genomic fabric paradigm can be applied to cardiovascular diseases, their significance is limited because of the heterocellular composition of the profiled left ventricle.

Through considering three independent groups of characteristics for each individual gene in each condition (illustrated in Figure 1), the GFP runs through the full potential of profiling tens of thousands of transcripts at a time on several biological replicas. The independence and complementarity of the three types of characteristics were proven previously for genes within the mTOR signaling pathway and evading apoptosis in human prostate [26,55], apoptosis in human thyroid [31], and chemokine signaling in human kidney [33]. They were also proven for chemokine signaling in mouse cortex [57], PI3K–AKT signaling in mouse hippocampus [58], and ion channels and transporters in mouse heart myocardia of each of the four chambers [6]. Regardless of the used high-throughput transcriptomic platform (RNA-sequencing, microarray, Affymetrix etc.), this strategy increased by 4–5 orders of magnitude the workable information furnished by a high-throughput transcriptomic study.

As one may note in Figure 1, the hypoxia not only changed the expression levels of the individual genes, but also their expression control and expression coordination in functional pathways. Substantial changes in the genes’ expression control and inter-coordination were constant findings in all previous GFP studies on diseases on samples collected from both humans [26,30,31,33,55] and animal models (e.g., [57,58,59,60]). We believe that such rich additional information could be instrumental in developing a personalized genomic medicine.

The GFP provides essential clues on the priorities of the cellular homeostatic mechanisms in controlling the expression of critical genes and especially how the genes are networked to optimize the functional pathways. That Lias is the most protected gene in the heart of 1-week normoxic mice did not come as surprise given the strong antioxidant potency of the α-lipoidic acid synthesized by the encoded enzyme [61]. Lias was significantly up-regulated (x = 1.52) in an ischemic heart but its expression was restored to normal by the cell treatment. Cadm4, the most protected gene in the heart of mice subjected during their first week of life to chronic intermittent hypoxia, is essential for restricting the production of cardiac outflow tract progenitor cells in zebrafish [62]. Whether it performs a similar function for the development of the mouse heart remains to be tested by further experiments. Finally, Numa1, the most protected gene after 1-week chronic constant hypoxia is a marker of the myotonic dystrophy type 1 [63]. As reported here and on previous genomic studies, disease [30,64,65,66], stress [11,23,67], and genetic manipulation [68,69] force the tissues to increase the control of the gene expression, presumably to limit the damages.

Importantly, GFP can hierarchize the genes according to their gene commanding height (GCH, Figure 2) that accounts for both the strength of the homeostatic controlling mechanisms of the expression accuracy and power to regulate the expression of other genes. With GCH, one can identify the gene master regulator of that condition (GMR) whose “smart” manipulation would have the desired effect on the cells it commands but little to no consequences on the other cells of the tissue. The monotonic relationship between the GCH and the transcriptomic consequences of altering the expression of that gene was proven by stable transfection of four genes into two standard human thyroid cancer cell lines [31,70].

Nonetheless, our approach improves quantification of the transcriptome alteration with a more accurate absolute fold-change cut-off to decide about the statistical significance of the expression regulation and especially with more comprehensive measures of individual gene contributions. Figure 3 makes a powerful case of the importance of adopting WIR and ITD for a better understanding of the transcriptomic consequences of a disease.

One of our primary focuses was on the key genes regulating myocardial adrenergic signaling, such as *Adra1b*, (Figure 4, Figure 5 and Figure 6), by means of the KEGG-built platform. The cardioprotective role of *Adra1b* has long been established by Woodcock’s group who first showed the reduction of reperfusion-induced Ins(1,4,5)P3 generation and arrhythmias in mouse hearts expressing constitutively active alpha1B-adrenergic receptors [71]. However, under pathologic conditions, e.g., pressure overload, overexpressing of alpha1B-adrenergic receptors leads to depressed contractile responses to beta-adrenergic receptor activation, and predisposes hearts to hypertrophy and worsen heart failure [72].

Our present study further revealed the remodeling of the *Adra1b* networking following myocardial infarction (group IN) and normalization by the post-myocardial infarction stem cell treatment (group IT) (Figure 6). The coordination analysis provided additional insights that would not be available through a simple comparison of expression levels of individual genes. For instance, *Crem* (CAMP-responsive element modulator) coordination with *Adra1b* was switched from synergistic in “NN” to antagonistic in “IN”, and practically independent in “IT”, indicating a major change in the interaction of the two genes. While the critical role of *Crem* in β-adrenoreceptor-mediated cardiac dysfunction is documented [73], there is not yet any report concerning the interaction between *Crem* and *Adra1b* and how such an interaction might impact the heart physiology. Since *Crem* is a transcription factor that mediates high-glucose response in cardiomyocytes [74], its relationship with Adra1b deserves further investigation.

## 5. Conclusions

So far, the GFP power to characterize the organization of the transcriptome was successfully tested in several other studies on neurological diseases (e.g., [35,57,58,59]), pulmonary hypertension [60], and cancers [26,30,31,33,55,70]. GFP proved also its ability to characterize the transcriptomic networks linking ionic channels and transporters across the heart chambers [6] as well as different cell types in insert systems [75]. The present report provides evidence of the advantages of using GFP analyses to decode the remodeling of the gene networks in the myocardial tissue following myocardial infraction or systemic hypoxia.

## Figures and Tables

**Figure 1 jpm-12-01246-f001:**
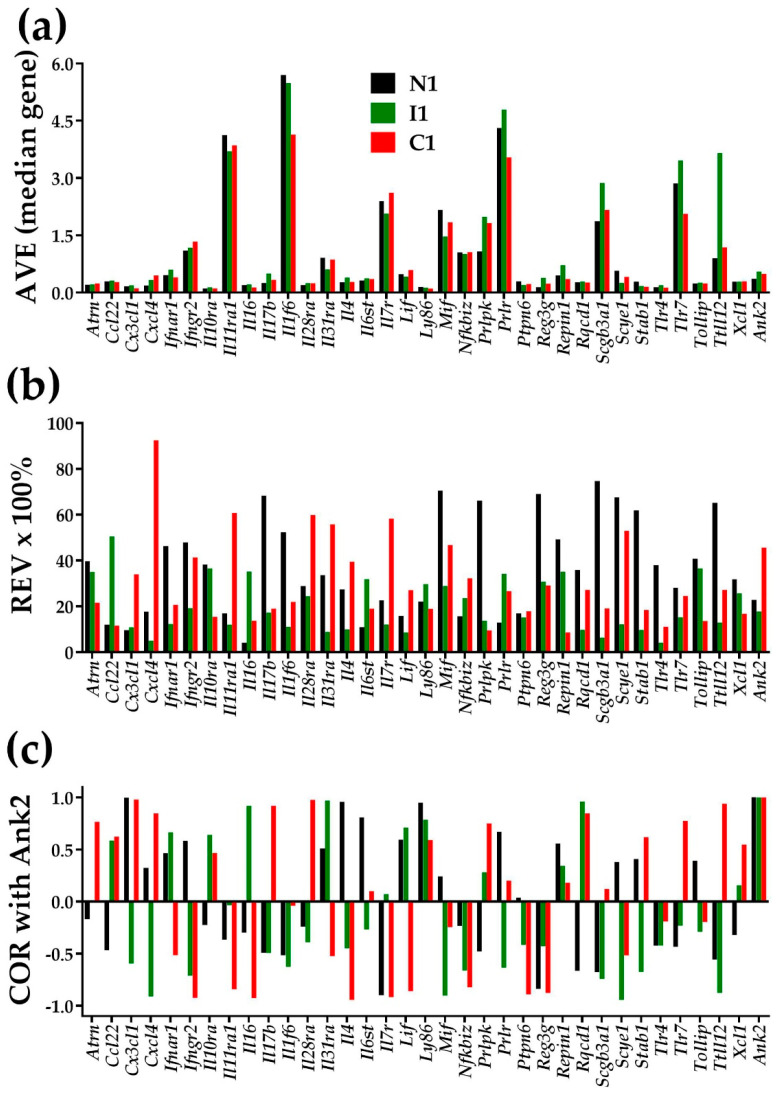
Illustration of the independence of the three types of characteristics of individual genes in the left ventricle of mice subjected in the first week of their life to normal atmospheric conditions (“N1”), chronic intermittent hypoxia (“I1”), or chronic constant hypoxia (“C1”). (**a**) Average expression level (AVE) in levels of the median gene in that condition. (**b**) Percentage of the relative expression variability (REV); (**c**) Expression correlation with *Ank2*.

**Figure 2 jpm-12-01246-f002:**
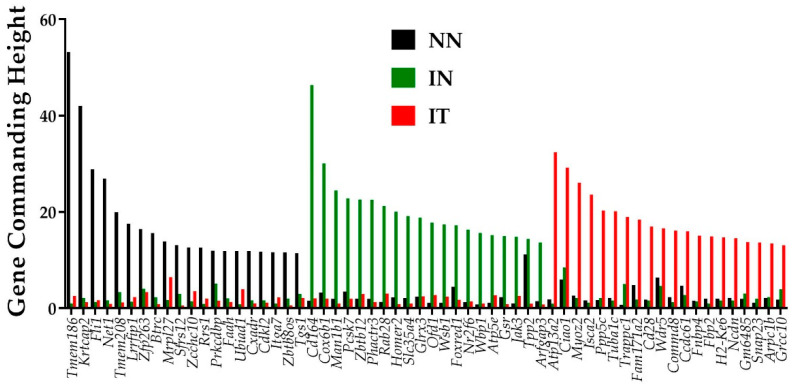
Top 20 genes in the conditions: normal untreated (“NN”), infarcted untreated (“IN”), infarcted treated (“IT”), normal untreated (“NN”), infarcted untreated (“IN”), and infarcted treated (“IT”). Note there is no overlap of the three gene sets and that the top genes in one condition have substantially lower GCH scores in the other two conditions.

**Figure 3 jpm-12-01246-f003:**
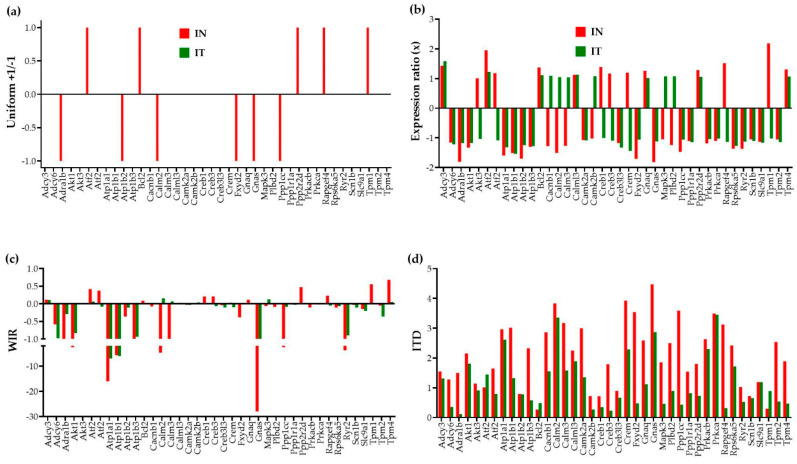
Four ways to report the altered expression of 40 individual genes involved in the adrenergic signaling in cardiomyocytes in untreated (“IN”) and treated (“IT”) post-ischemic infarcted mouse hearts with respect to healthy counterparts. (**a**) Uniform +1/−1 contribution of significantly up-/down-regulated genes. (**b**) Expression ratios of all genes. (**c**) Weighted individual (gene) regulation (WIR). (**d**) Individual (gene) transcriptomic distance (ITD).

**Figure 4 jpm-12-01246-f004:**
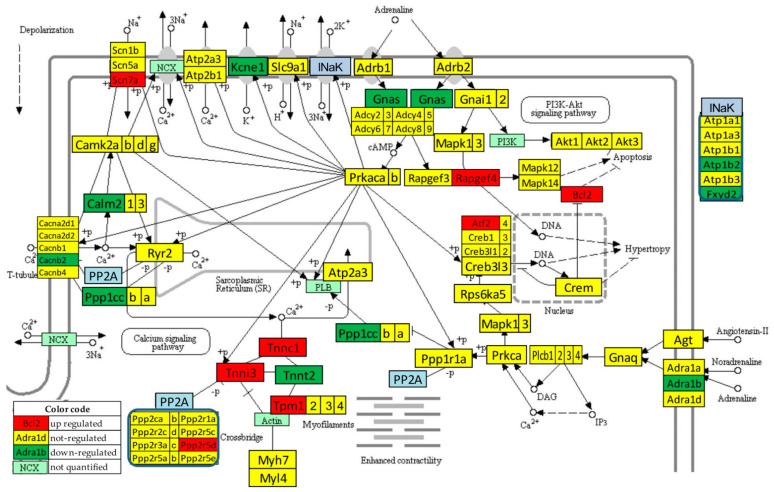
Statistically significant regulation of genes involved in the KEGG-built adrenergic signaling in the left ventricle cardiomyocytes of untreated mice with post-ischemic heart failure. INaK and PP2A are blocks of quantified genes, while NCX and PI3K are blocks of not quantified genes.

**Figure 5 jpm-12-01246-f005:**
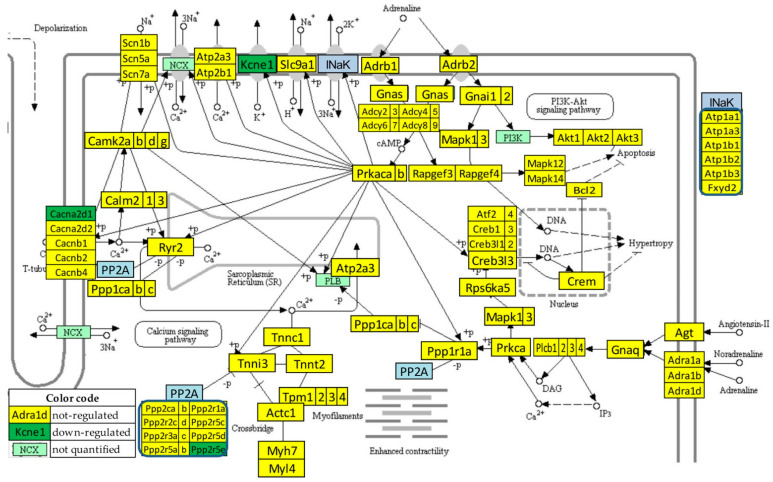
Statistically significant regulation of genes involved in the KEGG-built adrenergic signaling in the left ventricle cardiomyocytes of stem cell treated mice with post-ischemic heart failure. INaK and PP2A are blocks of quantified genes, while NCX and PI3K are blocks of not quantified genes.

**Figure 6 jpm-12-01246-f006:**
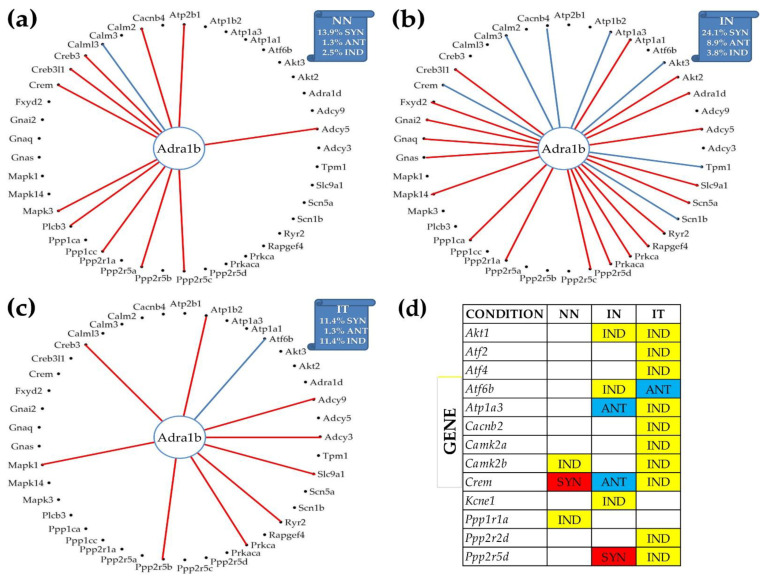
Remodeling of the *Adra1b* networking with genes from the KEGG-designed functional pathway “Adrenergic signaling in cardiomyocytes” caused by the post ischemic heart failure with and without stem cell treatment. (**a**) Significantly synergistically and antagonistically expressed partners of *Adra1b* in “NN” hearts. (**b**) Significantly synergistically and antagonistically expressed partners of *Adra1b* in “IN” hearts. (**c**) Significantly synergistically and antagonistically expressed partners of *Adra1b* in “IT” hearts. (**d**) Independently expressed genes with *Adra1b*.

**Table 1 jpm-12-01246-t001:** The 5 most controlled genes after 1, 2, and 4 weeks of life in normal (“N1”, “N2”, “N4”) atmospheric conditions, chronic intermittent (“I1”, “I2”, “I4”) and chronic constant (“C1”, “C2”, “C4”) hypoxia. Grey background of REV values indicates the most controlled 5 genes in each condition. For comparison, the average and the median REVs for all 9716 quantified genes are presented in each condition.

GENE	DESCRIPTION	N1	I1	C1	N2	I2	C2	N4	I4	C4
Lias	Lipoic acid synthetase	1.4	19.1	34.1	74.8	26.0	13.7	88.2	4.8	45.1
Psip1	PC4 and SFRS1 interacting protein 1	2.0	12.7	14.0	21.8	12.6	13.2	14.3	16.5	30.4
Ctdsp1	carboxy-terminal domain, RNA polymerase II, polypeptide A small phosphatase 1	2.1	45.5	61.0	72.1	24.1	13.4	24.2	16.2	16.9
Arhgef3	Rho guanine nucleotide exchange factor (GEF) 3	2.2	18.0	40.8	9.4	32.1	30.7	17.2	57.0	39.9
Rps6kb1	Ribosomal protein S6 kinase, polypeptide 1	2.4	30.9	16.2	105.9	44.6	12.0	65.2	37.0	43.5
Cadm4	cell adhesion molecule 4	57.5	0.1	4.8	59.7	39.5	27.4	76.8	13.0	40.8
Chfr	Checkpoint with forkhead and ring finger domains	39.5	1.1	20.0	21.7	6.3	5.2	15.4	22.1	22.2
Imp4	IMP4, U3 small nucleolar ribonucleoprotein, homolog (yeast)	23.5	1.2	19.0	99.1	47.9	26.9	23.9	14.9	30.9
Wdr63	WD repeat domain 63	29.2	1.4	3.8	46.5	10.9	10.8	59.6	10.3	7.8
Tpp1	Tripeptidyl peptidase I	45.9	1.7	23.0	30.5	52.8	27.4	22.2	35.0	25.7
Numa1	Nuclear mitotic apparatus protein 1	15.4	20.4	1.3	33.6	21.4	22.1	22.6	36.1	56.0
Sh3bp5	SH3-domain binding protein 5 (BTK-associated)	40.4	17.0	2.0	21.0	12.8	7.9	64.9	12.9	19.9
Rhbdf1	Rhomboid family 1 (Drosophila)	58.0	33.5	2.2	85.3	15.7	8.7	20.9	39.8	26.8
Pygb	Brain glycogen phosphorylase	34.4	20.4	2.2	23.8	61.6	34.7	50.1	47.4	34.1
Map3k7ip2	Mitogen-activated protein kinase kinase kinase 7 interacting protein 2	8.2	16.4	2.4	11.7	50.4	5.6	20.2	8.0	25.5
Ankrd15	Ankyrin repeat domain 15	33.5	19.6	11.1	1.0	53.3	18.9	47.0	6.2	16.0
Mrps5	Mitochondrial ribosomal protein S5	32.3	10.8	16.5	1.8	30.3	15.1	30.6	19.1	13.5
Gsdmdc1	Gasdermin domain containing 1	25.3	39.4	35.6	2.5	26.3	34.6	26.9	55.1	5.9
Pcdh7	Protocadherin 7	27.5	11.0	14.2	2.6	40.2	18.9	8.7	12.1	37.6
Gart	Phosphoribosylglycinamide formyltransferase	13.1	18.9	55.6	2.8	84.3	6.5	80.2	12.7	8.9
Tubg1	Tubulin, gamma 1	8.6	25.8	21.1	10.1	0.4	10.7	19.4	62.5	19.1
Zfp191	Zinc finger protein 191	12.1	8.9	6.9	24.7	1.2	9.4	11.9	25.5	31.7
Arhgef1	Rho guanine nucleotide exchange factor (GEF) 1	59.5	9.7	29.9	87.3	1.6	11.8	80.1	8.5	23.5
Bub3	Budding uninhibited by benzimidazoles 3 homolog (*S. cerevisiae*)	49.9	37.0	47.5	14.1	1.7	26.4	24.7	41.7	10.2
Gas2l1	Growth arrest-specific 2 like 1	57.0	37.0	41.8	22.4	1.7	43.2	23.9	14.8	14.9
Dmkn	Dermokine	25.9	18.9	58.2	34.9	11.6	1.1	11.9	14.3	57.3
Med6	Mediator of RNA polymerase II transcription, subunit 6 homolog (yeast)	31.5	11.1	7.8	68.1	23.1	1.2	21.4	10.8	37.1
Nt5dc1	5′-nucleotidase domain containing 1	22.3	29.0	18.3	36.8	11.2	1.5	47.3	9.3	32.2
Lima1	LIM domain and actin binding 1	10.9	24.2	45.5	6.8	24.3	1.5	71.5	16.8	16.9
Rpl27	Ribosomal protein L27	15.7	14.8	29.7	50.1	41.0	1.5	59.2	7.0	11.3
Mrpl15	Mitochondrial ribosomal protein L15	21.9	15.7	11.7	25.7	11.8	14.5	1.3	15.3	16.8
Agt	Angiotensinogen (serpin peptidase inhibitor, clade A, member 8)	36.7	32.3	21.8	7.9	53.7	23.2	2.7	25.3	34.3
Mxd1	MAX dimerization protein 1	26.4	28.2	24.1	43.5	20.6	40.8	3.0	60.8	14.2
Lama2	Laminin, alpha 2	15.5	20.7	23.5	5.9	37.7	9.3	3.1	29.5	8.4
Il31ra	Interleukin 31 receptor A	33.6	8.9	55.8	71.4	62.3	22.1	3.2	22.7	34.1
Qtrtd1	Queuine tRNA-ribosyltransferase domain containing 1	16.7	11.0	22.0	25.5	14.2	9.7	25.2	1.3	32.1
Gpbp1	GC-rich promoter binding protein 1	12.6	11.4	5.3	24.9	22.6	7.8	12.0	2.0	10.0
Hspb6	Heat shock protein, alpha-crystallin-related, B6	28.5	10.7	28.0	9.3	15.0	34.5	54.7	2.2	36.6
Rps13	Ribosomal protein S13	15.0	18.7	17.7	96.5	32.1	9.0	50.5	2.3	32.5
Sft2d3	SFT2 domain containing 3	27.8	15.5	18.1	21.8	30.6	25.5	21.9	2.3	24.6
Arid2	AT rich interactive domain 2 (Arid-rfx like)	17.6	7.1	18.6	44.2	37.4	8.9	37.9	12.0	0.6
Fos	FBJ osteosarcoma oncogene	8.8	34.6	21.1	17.1	25.4	13.1	41.1	16.3	1.6
Mlx	MAX-like protein X	38.0	8.2	23.5	40.3	56.5	14.8	22.0	25.9	2.1
Mrm1	Mitochondrial rRNA methyltransferase 1 homolog (*S. cerevisiae*)	49.2	18.5	13.3	30.0	24.0	24.3	31.5	50.7	2.4
Dynlrb1	Dynein light chain roadblock-type 1	24.8	44.2	12.8	40.8	29.9	17.2	19.1	17.2	2.4

**Average REV for the 9716 quantified genes**		30.5	22.6	27.9	47.2	31.5	20.5	37.5	25.0	33.0
**Median REV for the 9716 quantified genes**		27.0	19.8	24.8	40.5	28.3	18.7	33.6	23.3	30.9

## Data Availability

Experimental data used in this report were collected from the publicly accessible Gene Expression Omnibus (GEO) data bases: https://www.ncbi.nlm.nih.gov/sites/GDSbrowser?acc=GDS3655 (accessed on 10 June 2022), https://www.ncbi.nlm.nih.gov/geo/query/acc.cgi?acc=GSE29769 (accessed on 10 June 2022). https://www.ncbi.nlm.nih.gov/geo/query/acc.cgi?acc=GSE2271 (accessed on 10 June 2022).

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
