# Peer review of "Theory and Applications of the (Cardio) Genomic Fabric Approach to Post-Ischemic and Hypoxia-Induced Heart Failure"

_jpm, 2022, doi:10.3390/jpm12081246_

Round 1

Reviewer 1 Report

The manuscript "Theory and applications of the (cardio) genomic fabric approach to post-ischemic and hypoxia-induced heart failure" is very interesting. However, minor modifications are required to improve the manuscript quality. Overall, the manuscript describes the advantages of GFP analyses to decode the remodeling of the gene networks in the myocardial tissue during MI or systemic hypoxia.

Comments
Plagiarism check using Turnitin software shows more than a 20 % similarity index.
The methodology part is not well described.
Did you notice any sex difference in the results?

Include recent literature in the discussion part.

Author Response

Thank you so much for revising our work and kind appreciation. We apologize for the detected similarities in the description of our original methods with our previous publications. In order to avoid the similarities and fluidize the reading, part of the equations were moved to Appendix A and all sections have been re-written for clarity. All experimental groups were composed by two male and two female mice that did not allow a statistically significant assessment of sex differences. We have also updated and completed the cited Literature   

Reviewer 2 Report

In this manuscript, the authors present the mathematical theory behind genomic fabric paradigm (GFP), which takes in to account average expression levels of genes along with relative expression variability and the expression correlation with other genes. The authors apply this to microarray data from mouse models of ischemia and hypoxia and attempt to contrast GFP against traditional approaches to quantification of transcriptomic data. Overall, the manuscript is well written and data presented supports the authors' claims.

Author Response

Thank you so much for reviewing our work and kind appreciation.